# Learning NP-Hard Multi-Agent Assignment Planning using GNN: Inference on a Random Graph and Provable Auction-Fitted Q-learning

**Hyunwook Kang**[*]
Department of Computer Science
Texas A&M University
hwkang@tamu.edu

**Taehwan Kwon**
Kakao Brain
isaac.kwon@kakaobrain.com

**Jinkyoo Park**[†]
Industrial & Systems Engineering
KAIST
jinkyoo.park@kaist.ac.kr

**James R. Morrison**[†]
Electrical Engineering
Central Michigan University
morri1j@cmich.edu

## Abstract

This paper explores the possibility of near-optimally solving multi-agent, multi-task NP-hard planning problems with time-dependent rewards using a learning-based algorithm. In particular, we consider a class of robot/machine scheduling problems called the multi-robot reward collection problem (MRRC). Such MRRC problems well model ride-sharing, pickup-and-delivery, and a variety of related problems. In representing the MRRC problem as a sequential decision-making problem, we observe that each state can be represented as an extension of probabilistic graphical models (PGMs), which we refer to as random PGMs. We then develop a mean-field inference method for random PGMs. We then propose (1) an order-transferable Q-function estimator and (2) an order-transferability-enabled auction to select a joint assignment in polynomial-time. These result in a reinforcement learning framework with at least $1 - 1/e$ optimality. Experimental results on solving MRRC problems highlight the near-optimality and transferability of the proposed methods. We also consider identical parallel machine scheduling problems (IPMS) and minimax multiple traveling salesman problems (minimax-mTSP).

## 1 Introduction

**Motivation** Consider a set of identical robots seeking to serve a set of spatially distributed tasks. Each task is given an initial age (which then increases linearly in time). Greater rewards are given to younger tasks when service is complete according to a predetermined reward rule. Such problems prevail in operations research, e.g., dispatching drivers to transport customers or scheduling machines in a factory. Solving such highly structured NP-hard problems with the constraint of 'no possibility of two robots assigned to one task at once' using mathematical optimization schemes is infeasible or ineffective due to the expensive computational cost, especially when the problem size is large. Applying decentralized approach using multi-agent modeling frameworkLong et al. (2020); Rashid et al. (2018); Sunehag et al. (2017) is a possible way to solve a large scale problems. However, due to the impossibility of inducing consensus among agents in achieving the global objective without

---

[*]First author

[†]correspondence to: Jinkyoo Park (jinkyoo.park@kaist.ac.kr), James R. Morrison (morri1j@cmich.edu)

36th Conference on Neural Information Processing Systems (NeurIPS 2022).

effective communication Fischer et al. (1985), such decentralized approaches are rarely used in industries (e.g., factories). Thus, this study focuses on centralized methods for solving MRRC.

**Research Questions.** Many of such NP-hard scheduling problems have time-dependent rewards. To the best of our knowledge, these problems have not yet been addressed by non-decentralized learning-based methods. Even if one can, it must also be able to simultaneously address a fundamental challenge: the number of possible robot-task pairs to be considered increases exponentially. For instance, scheduling 8 robots and 50 tasks involves $10^{13}$ possible joint assignments at each time-step. The main research question that the current study seeks to resolve is "how to design a computationally effective (i.e., scalability in terms of learning and decision making) a learning based centralized decision making scheme for solving a large-scale NP-hard scheduling problems?"

**Proposed method and contributions.** The present paper explores the possibility of near-optimally solving multi-robot, multi-task NP-hard scheduling problems with time-dependent rewards using a learning-based algorithm. The study formulates multi-robot, multi-task NP-hard scheduling problems in a sequential decision-making framework and derives a joint scheduling policy with a theoretical performance bound under reasonable assumptions. The novelties of the current study are as follows:

- The study first observes that a state-joint assignment pair can be represented as a *random* PGM. After developing a theory of random PGM-based mean-field inference, we derive *random structure2vec*, a random PGM-based extension of structure2vec Dai et al. (2016).
- We estimate the $Q$-function $Q(s_k, a_k)$ using layers of *random structure2vec*, where $(s_k, a_k)$ is the state-joint action pair. Using an interpretation of a layer of structure2vec as a Weisfeiler-Lehman kernel (as in Dai et al. (2016)), we design the estimator to possess a property we call order-transferability. This property enables transferability in problem size.
- We propose a joint assignment rule called order-transferability-enabled auction policy (OTAP) to address exponential growth in joint assignment space. We propose auction-fitted $Q$-iteration (AFQI) by substitution of the `argmax` operation of fitted $Q$-iteration with OTAP to train the $Q$-function in a scalable manner. We prove that AFQI results in a policy with polynomial-time computation that achieves at least $1 - 1/e$ performance compared with the optimal policy.

**Results and Impacts.** Using simulation experiments, we show that the proposed policy typically achieves 97% optimality for the multi-robot reward collection (MRRC) problem in a deterministic environment with linearly time-varying rewards. This performance is well extended to experiments with stochastic traveling times. To the best of our knowledge, this result is the first to learn a near-optimal NP-hard multi-robot/machine scheduling policy with time-dependent rewards.

## 2   Related studies.

**Reinforcement Learning based Vehicle Routing Problems**. Mazyavkina et al. (2020) have categorized the RL approaches solving vehicle routing problems into two: (1) the improvement heuristics that learn an operator can iteratively improve the entire routing plans until there is no improvement is made (Wu et al., 2020; da Costa et al., 2020; Chen & Tian, 2019; Lu et al., 2020; Kim et al., 2021), (2) the construction heuristics that learn a policy that sequentially make a single routing action given the partial solution (state) (Bello et al., 2016; Nazari et al., 2018; Kool et al., 2018; Khalil et al., 2017), and (3) the hybrid approaches that mix these two approaches (Joshi et al., 2020; Fu et al., 2021; Kool et al., 2021; Ahn et al., 2020). These RL approaches have mainly considered a single-agent routing problem. Although these methods solve CVRP with multiple vehicles, they solve this problem from a single-agent perspective. In addition, most of these approaches consider the static reward function setting. On the contrary, our study explicitly considers multi-vehicle interaction while considering the time-varying reward, which is a more realistic routing problem setting.

**Graph Inference Based Approach**. Dai et al. (2017) showed that a graph neural network (GNN) called *structure2vec* Dai et al. (2016) can construct a solution for the Traveling Salesman Problem (TSP). *structure2vec* is a popular GNN derived from mean-field inference with a probabilistic graphical model (PGM). Dai et al. (2017) formulates the TSP as a Markov decision process (MDP) where a heuristically constructed PGM represents each state-next assignment pair. They employ *structure2vec* derived from the heuristic PGM to infer the $Q$-function, which they use to select the next assignment. While their choice of PGM was heuristic, their approach achieved near-optimality

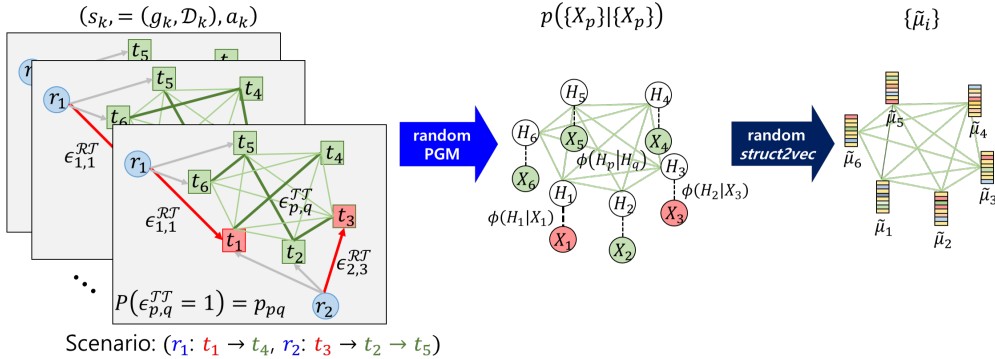

Figure 1: Representation of the scheduling problem using a random PGM

and transferability of their trained single-robot scheduling algorithm to new single-robot scheduling problems with an unseen number of tasks.

## 3 Multi-Robot Reward Collection Problem

In the main text, we model a general multi robot/machine scheduling problem as a discrete-time, discrete-state (DTDS) sequential decision-making problem. In the DTDS model, time advances in fixed increments $\Delta$, i.e., $t_k = t_0 + \Delta \times k$ where $t_k$ is the actual time after $k$ decision epochs have passed. For simplicity, we use $k$ as a time index representing the $k$th decision epoch. In this framework, $s_k$ denotes a state, and action $a_k$ denotes a joint assignment of robots/machines to unfinished tasks at the $k$th epoch. The objective of the problem is to learn the optimum scheduling policy $\pi_\theta : s_k \to a_k$ that maximizes the reward collected or minimizes the total completion time (a.k.a. makespan minimization). Below is the formulation of MRRC. We additionally propose a continuous-time continuous-state (CTCS) problem to identical parallel machine scheduling problems (IPMS) in Appendix A and minimax multiple traveling salesman problems (minimax-mTSP) in Appendix B.

### 3.1 State

The state $s_k$ at epoch $k$ is represented as $(g_k, \mathcal{D}_k)$ where a graph $g_k = ((\mathcal{R}, \mathcal{T}_k), (\mathcal{E}_k^{\mathcal{T}\mathcal{T}}, \mathcal{E}_k^{\mathcal{R}\mathcal{T}}))$ and associated feature set $\mathcal{D}_k = (\mathcal{D}_k^{\mathcal{R}}, \mathcal{D}_k^{\mathcal{T}}, \mathcal{D}_k^{\mathcal{T}\mathcal{T}}, \mathcal{D}_k^{\mathcal{R}\mathcal{T}})$. The elements of graph $g_k$ are defined as (See Figure 1):

- $\mathcal{R} = \{1, ..., M\}$ is the index set of all robots. The index $i$ and $j$ will be used to specifically denote robots.
- $\mathcal{T}_k = \{1, ..., N\}$ is the index set of all remaining unserved tasks at decision epoch $k$. The index $p$ and $q$ will be used to specifically denote tasks.
- $\mathcal{E}_k^{\mathcal{T}\mathcal{T}} = \{\epsilon_{pq}^{\mathcal{T}\mathcal{T}} | p \in \mathcal{T}_k, q \in \mathcal{T}_k\}$ is the set of all directed edges from a task in $\mathcal{T}_k$ to any task in $\mathcal{T}_k$. We consider each edge as a random variable. The task-to-task edge $\epsilon_{pq}^{\mathcal{T}\mathcal{T}} = 1$ indicates the event that a robot that has just completed task $p$ subsequently completes task $q$. We denote the probability $p(\epsilon_{pq}^{\mathcal{T}\mathcal{T}} = 1) \in [0, 1]$ the presence probability of the edge $\epsilon_{pq}^{\mathcal{T}\mathcal{T}}$.
- $\mathcal{E}_k^{\mathcal{R}\mathcal{T}} = \{\epsilon_{ip}^{\mathcal{R}\mathcal{T}} | i \in \mathcal{R}, p \in \mathcal{T}_k\}$ is the set of all directed edges from a robot in $\mathcal{R}$ to a task in $\mathcal{T}_k$. We say the robot-to-task edge $\epsilon_{ip}^{\mathcal{R}\mathcal{T}} = 1$ indicates the event that robot $i$ is assigned to the task $p$. This edge is defined deterministically depending on the joint assignment action. If robot $i$ is assigned to task $p$, then $p(\epsilon_{ip}^{\mathcal{R}\mathcal{T}}) = 1$, otherwise 0.

The element of feature set $\mathcal{D}_k$ associated with the graph $g_k$ is defined as:

- $\mathcal{D}_k^{\mathcal{R}} = \{d_i^{\mathcal{R}} | i \in \mathcal{R}\}$ is the set of node features for the robot nodes in $\mathcal{R}$ at epoch $k$. In MRRC, $d_i^{\mathcal{R}}$ is defined as the location of robot $i$ at epoch $k$ (epoch index $k$ is omitted).
- $\mathcal{D}_k^{\mathcal{T}} = \{d_p^{\mathcal{T}} | p \in \mathcal{T}_k\}$ is the set of node features for the task nodes in $\mathcal{T}_k$ at epoch $k$. In MRRC, $d_p^{\mathcal{T}}$ is defined as the age of task $p$ at epoch $k$ (epoch index $k$ is omitted).

- $\mathcal{D}_k^{\mathcal{TT}} = \{d_{pq}^{\mathcal{TT}} | p \in \mathcal{T}_k, q \in \mathcal{T}_k\}$ is the set of task-task edge features at epoch $k$. $d_{pq}^{\mathcal{TT}}$ denotes the duration for a robot that has just completed task $p$ to subsequently compete task $q$. We call this duration *task completion time*. In MRRC, a task completion time is given as a random variable (in practice, our method only requires a set of samples of random variable).
- $\mathcal{D}_k^{\mathcal{RT}} = \{d_{ip}^{\mathcal{RT}} | i \in \mathcal{R}, p \in \mathcal{T}_k\}$ is the set of robot-task edge features at epoch $k$. $d_{ip}^{\mathcal{RT}}$ denotes the traveling time for robot $i$ to reach task $p$.

## 3.2 Action

An action $a_k$, a joint assignment at epoch $k$, is defined as a maximal bipartite matching of the complete bipartite graph $(\mathcal{R}, \mathcal{T}_k, \mathcal{E}_k^{\mathcal{RT}})$ composed of the robot nodes $\mathcal{R}$, the remaining task nodes $\mathcal{T}_k$, and the fully connected edges between them $\mathcal{E}_k^{\mathcal{RT}}$. That is, given the current state $s_k = (g_k, \mathcal{D}_k)$, $a_k$ is a subset of $\mathcal{E}_k^{\mathcal{RT}}$ satisfying (1) no two robots can be assigned to the same tasks, and (ii) a robot may only remain without assignment when the number of robots exceeds the number of remaining tasks. If $\epsilon_{ip}^{\mathcal{RT}} \in a_k$, it means that robot $i$ is assigned with task $p$ at epoch $k$. For example, Figure 1 shows the case where $a_k = (\epsilon_{1,1}^{\mathcal{RT}}, \epsilon_{2,3}^{\mathcal{RT}})$. (Note, we equivalently may say this as $\epsilon_{1,1}^{\mathcal{RT}} = \epsilon_{2,3}^{\mathcal{RT}} = 1$ and 0 otherwise.) In MRRC, all robots are allowed to change their assignments at each decision epoch. (In IPMS and mTSP, only free machines/salesmen are newly assigned.)

## 3.3 State transition

As the joint assignment $a_k$ is executed given the current state $s_k = (g_k, \mathcal{D}_k)$, the next state $s_{k+1} = (g_{k+1}, \mathcal{D}_{k+1})$ is determined with the updated graph $g_{k+1}$ and features $\mathcal{D}_{k+1}$.

**Graph update.** When the decision epoch corresponds to the point when task $p$ is completed, the corresponding task node will be removed in the updated task nodes as $\mathcal{T}_{k+1} = \mathcal{T}_k/\{p\}$, and the task-task edges and robot-task edges, $\mathcal{E}_{k+1}^{\mathcal{TT}}$ and $\mathcal{E}_{k+1}^{\mathcal{RT}}$, will be accordingly updated.

**Feature update.** At decision epoch $k + 1$, $\mathcal{D}_{k+1} = (\mathcal{D}_{k+1}^{\mathcal{R}}, \mathcal{D}_{k+1}^{\mathcal{T}}, \mathcal{D}_{k+1}^{\mathcal{TT}}, \mathcal{D}_{k+1}^{\mathcal{RT}})$ is determined. In MRRC, task locations $\mathcal{D}_{k+1}^{\mathcal{R}} = \{d_i^{\mathcal{R}} | i \in \mathcal{R}\}$ and task ages $\mathcal{D}_{k+1}^{\mathcal{T}} = \{d_p^{\mathcal{T}} | p \in \mathcal{T}_{k+1}\}$ are updated. The robot-task edge features $\mathcal{D}_{k+1}^{\mathcal{RT}}$ will be updated according to $\mathcal{D}_{k+1}^{\mathcal{R}}$ as well.

## 3.4 Reward and objective

At time 0, each task is given an initial age which increases linearly in time. A reward $r_k = r\left(d_p^{\mathcal{T}}\right)$ is given when a task $p \in \mathcal{T}_k$, whose age is $d_p^{\mathcal{T}}$, is served at epoch $k$. We consider linear and nonlinear reward functions $r$ for MRRC. The objective is to learn a stationary policy $\pi$, a function that maps current state $s$ into current action $a$, to maximize expected total collected rewards $Q^\pi(s, a) =: E_{P,\pi}[\sum_{k=0}^{\infty} R(s_{t_k}, a_{t_k}, s_{t_{k+1}}) | s_{t_0} = s, a_{t_0} = a]$.

# 4 Random graph embedding: *RandStructure2Vec*

We observe that when task completion time is not deterministic, a scheduling problem can be represented as an extension of probabilistic graphical models (PGMs), which we refer to as random PGMs. This section proposes a mean-field inference method, *random struct2vec*, for random PGMs to estimate the state-action value $Q(s_k, a_k)$ in solving MRRC.

## 4.1 Random PGM for representing a state of MRRC

Given random variables $\mathcal{X} = \{X_p\}$, suppose that we can factor the joint distribution $p(\mathcal{X})$ as $p(\mathcal{X}) = \frac{1}{Z} \prod_i \phi_i(\mathcal{D}_i)$ where $\phi_i(\mathcal{D}_i)$ denotes a marginal distribution or conditional distribution associated with a set of random variables $\mathcal{D}_i$; $Z$ is a normalizing constant. Then $\{X_p\}$ is called a probabilistic graphical model (PGM). In a PGM, $\mathcal{D}_i$ is called a clique and $\phi_i(\mathcal{D}_i)$ is called the clique potential for $\mathcal{D}_i$, and $\mathcal{D}_i$ is called the scope of $\phi_i$. We often write simply $\phi_i$, suppressing $\mathcal{D}_i$.

Starting from a state $s_k$ and an action $a_k$, one can conduct a random experiment of "sequential decision making using policy $\pi$". In this random experiment, we can denote the events 'How robots serve all remaining tasks in which sequence' as *scenarios*. For example, suppose that at time-step $k$ we

are given robots $\{r_1, r_2\}$, tasks $\{t_1, t_2, t_3, t_4, t_5, t_6\}$ and we follow the policy $\pi$ onward. One possible scenario is that robot $r_1$ serves tasks $\{t_1 \to t_4\}$ and robot $r_2$ serves tasks $\{t_3 \to t_2 \to t_5 \to t_6\}$ (see Figure 1). Note that the time when $t_5$ is served depends on the time when $t_2$ is served (state transition is Markovian); thus the reward from $t_5$ depends on the reward from $t_2$. As shown in Figure 1, $\{\{t_1 \to t_4\}, \{t_3 \to t_2 \to t_5 \to t_6\}\}$ can be represented as a single instance of a Bayesian Network. Since scenario realization is random, we can construct the distribution over such scenarios using 'random' Bayesian Network with random node $X_k = (s_k, a_k)$ and clique potential $\phi$. For details, see section 5.1.

## 4.2 Mean-field inference with random PGM

Let $\mathcal{X} = \{X_p\}$ be the set of all random variables in the inference problem. Let $\mathcal{G}_{\mathcal{X}}$ be the set of all possible PGMs on $\mathcal{X}$. Let $\mathcal{P} : \mathcal{G}_{\mathcal{X}} \mapsto [0, 1]$ be a probability measure on $\mathcal{G}_{\mathcal{X}}$. Define a random PGM on $\mathcal{X}$ as $\{\mathcal{G}_{\mathcal{X}}, \mathcal{P}\}$. Note that the inference of $\{\mathcal{G}_{\mathcal{X}}, \mathcal{P}\}$ will be difficult; $|\mathcal{G}_{\mathcal{X}}|$ is too large for inferring $\mathcal{P}$ even using Monte-Carlo sampling approach. To avoid this difficulty, we use the approximated inference using *semi-cliques*. Suppose that we are given the set of all possible cliques on $\mathcal{X}$ as $\mathfrak{C}_{\mathcal{X}}$. As a PGM will be realized according to $\mathcal{P}$, only a few of the possible cliques in $\mathfrak{C}_{\mathcal{X}}$ will be actually realized as an element of the PGM and become real cliques. We call such potential clique elements of $\mathfrak{C}_{\mathcal{X}}$ as *semi-cliques*. Note that if we were given $\mathcal{P}$, we could calculate the presence probability $p_m$ of the semi-clique $\mathcal{D}_m$ as $p_m = \sum_{G \in \mathcal{G}_{\mathcal{X}}} \mathcal{P}(G) \mathbf{1}_{\mathcal{D}_m \in G}$, where $\mathbf{1}$ denotes the indicator function.

**Mean-field inference with random PGM.** We start from a specific inference problem and state the main theorem in more general way. Consider a random PGM on $\mathcal{X} = (\{H_i\}, \{X_j\})$ where $H_k$ is the latent variable corresponding to the observed variable $X_k$. Our goal is to infer $\{H_i\}$ given $\{X_j\}$ by finding $p(\{H_i\}|\{x_j\})$. In mean-field inference, we instead find a set of surrogate distributions $\{q^{\{x_j\}}(H_i)\}$ for which $\{H_i\}$ are independent. Here $q^{\{x_j\}}$ means that $q$ is a function of $\{x_j\}$).

We next state ***Theorem 1.*** in a very general manner. The statement is the same as that of mean-field inference with PGM Koller & Friedman (2009) except that ours has the presence probability terms $\{p_m\}$ of semi-cliques $\{\mathcal{D}_m\}$. The implication is that inference of presence probability of each semi-clique is enough to conduct mean-field inference, and the inference of $\{\mathcal{G}_{\mathcal{X}}, \mathcal{P}\}$ is not needed.

***Theorem 1****. Random PGM based mean field inference. Suppose a random PGM on $\mathcal{X} = \{X_p\}$ is given, and the presence probability $\{p_m\}$ for all semi-cliques $\mathcal{D}_m \in \mathfrak{C}_{\mathcal{X}}$ are known. Then, the surrogate distribution $\{q_p(x_p)\}$ in mean-field inference is optimal only if $q_p(x_p) = \frac{1}{Z_p} \exp\left\{ \sum_{m:X_p \in \mathcal{D}_m} p_m \mathbb{E}_{(\mathcal{D}_m - \{X_p\}) \sim q} [\ln \phi_m(\mathcal{D}_m, x_p)] \right\}$ where $Z_p$ is a normalizer and $\phi_m$ clique potential for $\mathcal{D}_m$.*

For the general background and proof, see Appendix C.

**RandStructure2Vec.** In Dai et al. (2016), structure2vec was derived as a vector space embedding of mean-field inference with PGM. For detailed background on vector space embedding, see Appendix D. From ***Theorem 1***, we derive *random structure2vec* as a vector-space embedding of mean-field inference with random PGM. Suppose that once a PGM is realized the PGM has joint distribution proportional to some factorization $\prod_p \phi(H_p|I_p) \prod_{p,q} \phi(H_p|H_q)$ (as in Dai et al. (2016)). Under this assumption, we can write $\{q^{\{x_j\}}(H_i)\}$ as $\{q^{x_i}(H_i)\}$. In Dai et al. (2016), they suggest that structure2vec is essentially a fixed point iteration $\tilde{\mu}_p \leftarrow \sigma(W_1 x_p + W_2 \sum_{q \neq p} \tilde{\mu}_q)$ where $\tilde{\mu}_p$ is a latent vector for node $p$ and $x_p$ is input for node $p$. They show that, when we interpret $\tilde{\mu}_i$ as a vector space injective embedding expressed as $\tilde{\mu}_i = \int_{\mathcal{H}} \phi(h_i) q^{x_i}(h_i) dh_i$ for some $\phi$, structure2vec's fixed point iteration is the embedding of fixed point iteration of mean-field inference with the PGM. *Lemma 1* states that we have a similar result for random PGM by including only the $\{p_{qp}\}$ information.

***Lemma 1****. Structure2vec for random PGM. Assume that the presence probabilities $\{p_{qp}\}$ for all pairwise semi-cliques $\mathcal{D}_{qp} \in \mathfrak{C}_{\mathcal{X}}$ are given. Then embedding the fixed point equation in Theorem 1 generates the fixed point equation $\tilde{\mu}_p \leftarrow \sigma\left(W_1 x_p + W_2 \sum_{q \neq p} p_{qp} \tilde{\mu}_q\right)$. We refer to this fixed point iteration as random structure2vec. (The proof of Lemma 1 can be found in Appendix E.)*

**Remarks.** Note that inference of $\{\mathcal{G}_\mathcal{X}, \mathcal{P}\}$ is in general a difficult task. One implication of *Theorem 1* is that *we transformed a difficult inference task into a simple inference task*: inferring the presence probability of each semi-clique. (See Appendix F for the algorithm that conducts this task.) In addition, *Lemma 1* provides a theoretical justification to ignore the inter-dependencies among edge presences when embedding a random graph using GNN. When graph edges are not explicitly given or known to be random, the simplest heuristic one can use is to separately infer the presence probabilities of all edges and adjust the weights of GNN's message propagation. According to *Lemma 1*, possible inter-dependencies among edges would not affect the quality of such heuristic inference.

## 5    Solving MRRC with *RandStructure2Vec*

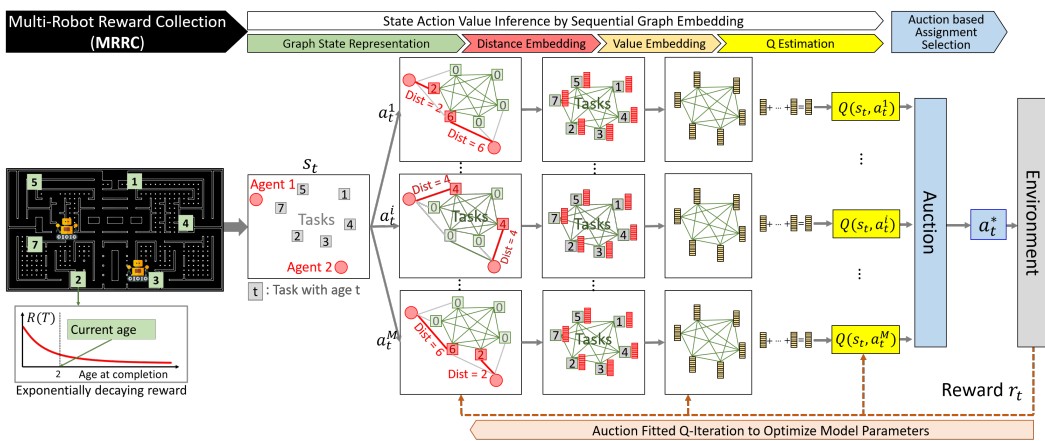

Figure 2: State representation and main inference procedure

This section describes how the proposed method solves the MRRC problem with the newly developed *random structure2vec* method. The total solution (i.e., the sequential assignments of robots/machines to tasks) is found by iteratively repeating a sequential decision making. This section specifically describes how to choose a joint assignment $a_k$ given the current state $s_k = (g_k, \mathcal{D}_k)$. The procedure for determining a joint assignment action is composed of (1) represent the state using a random Bayesian Network, (2) estimate the Q-value using random graph embedding, and (3) select a joint assignment. Figure 2 depicts the overall procedure. This section focuses on MRRC; the appendices will provide the formulation, solution procedure, and results for IPMS and mTSP problems as well.

### 5.1    Representing a state using a random PGM

In this section, we describe how the MRRC problem state $s_k$ and one possible joint assignment $a_k$ can be represented as a Bayesian network. Assume $H_p$, the hidden random variable for task $p$, carries information about the benefit of serving task $p$. Given a scenario (see section 3.1), $H_p$ depends on $H_q$ only if $q$ is served after $p$ by the same robot, and $H_p$ depends on $X_p$, the observed features for task $p$. For an MRRC problem, $X_p$ can be either a task's assignment information $d_{ip}^{\mathcal{R}\mathcal{T}}$ or a task's age $d_p^{\mathcal{T}}$. Based on these definitions, a Bayesian Network for $\{H_p\}$ and $\{X_p\}$ is constructed as $p(\{H_p\}|\{X_p\}) = \prod_p \phi(H_p|X_p) \prod_{p,q} \phi(H_p|H_q)$. This Bayesian Network corresponds to one scenario; there are many possible scenarios that can be realized randomly. Since random PGM naturally models this characteristic, according to Lemma 1, we are justified to use *random structure2vec* with edge (semi-clique) presence probabilities $\{p(\epsilon_{pq}^{\mathcal{T}\mathcal{T}})\}$ from in section 2.1.

### 5.2    Estimating state-action value using Order trainability-enabled Q-function

We illustrate how we can estimate a Q-function for MRRC by applying *random structure2vec* to the random PGM that represents $(s_k, a_k)$. *Lemma 1* provides the fixed point equation $\tilde{\mu}_p = \sigma\left(W_1 x_p + W_2 \sum_{j \neq k} p_{qp} \tilde{\mu}_q\right)$ to compute the embeddings $\tilde{\mu}_p$ for node task $p$ in a random PGM. The embeddings $\tilde{\mu}_p$ for $p \in \mathcal{T}_k$ are computed in an iterative manner using *random structure2vec* as:

$\tilde{\mu}_p^{(\tau+1)} = \sigma\left(W_1 x_p + W_2 \sum_{q \neq p} p_{qp} \tilde{\mu}_q^{(\tau)}\right)$. We propose a network architecture composed of two-step sequential *random structure2vec*, which is composed of *action embedding* and *value embedding*.

**Action Embedding.** The first *random structure2vec* layer embeds the joint assignment into the task nodes $\mathcal{T}_k = \{1, ..., n\}$. The action embedding $\tilde{\mu}_p^A$ for task node $p$ is defined as the fixed point for the equation $\tilde{\mu}_p^A = \sigma\left(W_1^A x_p^A + W_2^A \sum_{q \neq p} p_{qp} \tilde{\mu}_q^A\right)$ where $x_p^A = d_{ip}^{\mathcal{RT}}$ (distance from robot $i$ to task $p$) when task $p$ is assigned to robot $i$, $x_p^A = 0$ when task $p$ is not assigned. The action node embeddings $\{\tilde{\mu}_p^A | p \in \mathcal{T}_k\}$, computed via iteration simultaneously by employing *random structure2vec*, provide sufficient information about the relative locations between robots and their assigned tasks.

**Value Embedding.** The second *random structure2vec* layer embeds the task ages into the task nodes. The value embedding $\tilde{\mu}_p^V$ for task node $p$ is defined as the fixed point for the equation $\tilde{\mu}_p^V = \sigma\left(W_1^V x_p^V + W_2^V \sum_{q \neq p} p_{qp} \tilde{\mu}_q^V\right)$ where $x_p^V = (\tilde{\mu}_p^A, d_p^{\mathcal{T}})$ is the concatenation of the action embedding $\tilde{\mu}_p^A$ computed by the first layer and the task age $d_p^{\mathcal{T}}$ for node $p$. The resulting value node embeddings $\{\tilde{\mu}_p^V | p \in \mathcal{T}_k\}$, computed in an iterative manner, provide sufficient information about how much value is likely in the local graph around each task by the specified joint assignment.

**Computing** $Q_\theta(s_k, a_k)$. To derive $Q_\theta(s_k, a_k)$, we aggregate the embedding vectors for all nodes by $\tilde{\mu}^V = \sum_p \tilde{\mu}_p^V$ to obtain one global vector $\tilde{\mu}^V$ to embed the value affinity of the global graph. We then use a neural network to map $\tilde{\mu}^V$ into $Q_\theta(s_k, a_k)$. The overall pseudo-code for estimating steps is provided in Appendix G.

It is essential for the proposed inference method can estimate the state action value $Q_\theta(s_k, a_k)$ for varying graph size $g_k$. Let us provide the intuition related to scuh problem-size transferability. For *Action Embedding*, transferability is trivial; the inference problem is a scale-free task *locally around each node*. For *Value Embedding*, consider the ratio of robots to tasks. The overall value affinity embedding will be underestimated if this ratio in the training environment is smaller than this ratio in the testing environment; overestimated overall otherwise. The intuition is that this over/under-estimation does not matter in Q-function based policies as discussed in van Hasselt et al. (2015) as long as the *order* of Q-function value among actions are the same. That is, as long as the best assignments chosen are the same, i.e., $\arg\max_{a_k} Q(s_k, a_k) = \arg\max_{a_k} Q_\theta(s_k, a_k)$, the magnitude of imprecision $|Q(s_k, a_k) - Q_\theta(s_k, a_k)|$ does not matter. We call this property of an estimator *order-transferability* (with respect to the max operation).

## 5.3 Selecting a joint assignment using OTAP

With the previously introduced way to estimate $Q_\theta(s_k, a_k)$, we illustrate how to compute the joint assignment (action) $a_k^*$, a maximal bipartite matching in the bipartite graph $(\mathcal{R}, \mathcal{T}_k, \mathcal{E}_k^{\mathcal{RT}})$, given the state $s_k = (g_k, D_k)$. Specifically, we propose the order transferability-enabled auction policy (OTAP) that constructs a joint assignment $a_k$ through $N = \max(|\mathcal{R}|, |k|)$ iterations of *Bidding* and *Consensus* phases. This auction follows the spirit of Sequential Single Item (SSI) auctioning (Koenig et al. (2006)); each iteration adds one robot-task assignment to construct a full joint assignment.

**Bidding-phase.** In the $n^{th}$ iteration of the bidding phase, given $\mathcal{M}_\theta^{(n-1)}$, the ordered set of $n-1$ robot-task edges in $\mathcal{E}_k^{RT}$ determined by the previous $n-1$ iterations, all the unassigned robots bid their the most preferable task to conduct. Respecting robot-task assignments determined in previous $n-1$ iterations and ignoring other unassigned robots, each unassigned robot $i$ select the best task assignment $\epsilon_{il}^{\mathcal{RT}}$ that maximizes $Q_\theta^n(s_k, \mathcal{M}_\theta^{(n-1)} \cup \{\epsilon_{ip}^{\mathcal{RT}}\})$ among all unassigned tasks $p \in \mathcal{T}_k$, where $Q_\theta^n$ is the $\theta$-parameterized network with superscript $n$ indicating that the state action value is estimated at $n^{th}$ iteration. Then, robot $i$ bids $\{\epsilon_{i\ell}^{\mathcal{RT}}, Q_\theta^n(s_k, \mathcal{M}_\theta^{(n-1)} \cup \{\epsilon_{i\ell}^{\mathcal{RT}}\})\}$ to the auctioneer. This bidding occurs simultaneously by all the unassigned robots at the $n^{th}$ iteration. Since the number of ignored robots varies at each iteration, transferability of Q-function inference is crucial.

**Consensus-phase.** In the consensus phase of $n^{th}$ iteration, the centralized auctioneer finds the bid with the best bid value, say $\{\epsilon_{i^*p^*}^{\mathcal{RT}}, Q_\theta^n(s_k, \mathcal{M}_\theta^{(n-1)} \cup \{\epsilon_{i^*p^*}^{\mathcal{RT}}\})\}$ (Here $i^*$ and $p^*$ denote the best robot task pair.) Denote $\epsilon_{i^*p^*}^{\mathcal{RT}} := m_\theta^{(n)}$. The auctioneer sets everyone's $\mathcal{M}_\theta^{(n)}$ as $\mathcal{M}_\theta^{(n)} = \mathcal{M}_\theta^{(n-1)} \cup m_\theta^{(n)}$ and initiate bidding phase for the remaining unassigned robots.

These two phases iterate until reaching $\mathcal{M}_\theta^{(N)} = \{m_\theta^{(1)}, \ldots, m_\theta^{(N)}\}$. This $\mathcal{M}_\theta^{(N)}$ is chosen as the joint assignment $a_k^*$ of $N$-robots at time step $k$. That is, $\pi_\theta(s_k) = a_k^*$. The computational complexity for computing $\pi_\theta$ is $\mathrm{O}\left(|R| \, |T|\right)$ and is only polynomial; see Appendix L.

### 5.4 Training Q-function using AFQI

The fitted Q-iteration (FQI) finds $\theta$ that minimizes $E_{(s_k, a_k, r_k, s_{k+1}) \sim D}\left[Q_\theta\left(s_k, a_k\right) - [r\left(s_k, a_k\right) + \gamma \max_a Q_\theta\left(s_{k+1}, a\right)]\right]$ where $D$ denotes the distribution of training data. We propose a new rein­forcement learning method, which we call *Auction-fitted Q-iteration (AFQI)*, which replaces $\max_a$ $Q_\theta\left(s_k, a\right)$ used in the conventional FQI with OTAP. That is, writing OTAP as $\pi_{Q_\theta}$, AFQI finds $\theta$ that empirically minimizes $E_{(s_k, a_k, r_k, s_{k+1}) \sim D}\left[Q_\theta\left(s_k, a_k\right) - [r\left(s_k, a_k\right) + \gamma Q_\theta\left(s_{k+1}, \pi_{Q_\theta}\left(s_{k+1}\right)\right)]\right]$.

In learning the parameters $\theta$ for $Q_\theta\left(s_k, a_k\right)$, we use the exploration strategy that perturbs the parameters $\theta$ randomly to actively explore the joint assignment space with OTAP. While this method was originally developed for policy-gradient based methods Plappert et al. (2017), exploration in parameter space is useful in our auction-fitted Q-iteration since it generates a reasonable combination of assignments.

## 6 Theoretical analysis

We show the proposed AFQI obtains at least $1 - 1/e$ optimality and enables computation of the joint assignment in polynomial time. This result is achieved by the order-transferability of the proposed $Q$-function estimator and its use in selecting the joint assignment.

### 6.1 Performance bound of OTAP

Recall that $Q^n$ denotes the $n$-robot problem's true $Q$-function. In the same way as we defined $\mathcal{M}_\theta^{(N)}$ above, denote the joint assignment chosen by OTAP as $\{Q^n\}_{n=1}^N$ as $\mathcal{M}^{(N)} = \{m^{(1)}, \ldots, m^{(N)}\}$.

**_Lemma 2._** *If the $Q$-function approximator has order transferability, then $\mathcal{M}^{(N)} = \mathcal{M}_\theta^{(N)}$.*

For any decision epoch $k$, let $\mathcal{M}$ denote a set of robot-task pairs (a subset of $\mathcal{E}_k^{\mathcal{RT}}$). For any robot-task pair $m \in \mathcal{E}_k^{\mathcal{RT}}$, define $\Delta(m \mid \mathcal{M}) := Q^{|\mathcal{M} \cup \{m\}|}(s_k, \mathcal{M} \cup \{m\}) - Q^{|\mathcal{M}|}(s_k, \mathcal{M})$ as the the marginal value (under the true $Q$-functions) of adding robot-task pair $m \in \mathcal{E}_k^{\mathcal{RT}}$. Lemma 2 enables us to use the result discussed in Nemhauser et al. (1978) and achieve the result of Theorem 2.

**_Theorem 2._** *Suppose that the $Q$-function approximation with the parameter value $\theta$ exhibits order transferability. Denote $\mathcal{M}_\theta^{(N)}$ as the result of OTAP using $\{Q_\theta^n\}_{n=1}^N$ and let $\mathcal{M}^* = \mathrm{argmax}_{a_k}$ $Q^{|a_k|}(s_k, a_k)$. If $\Delta(m \mid \mathcal{M}) \geq 0, \forall \mathcal{M} \subset \mathcal{E}_k^{RT}, \forall m \in \mathcal{E}_k^{\mathcal{RT}}$, and the marginal value of adding one robot diminishes as the number of robots increases, i.e., $\Delta(m \mid \mathcal{M}) \leq \Delta(m \mid \mathcal{N}), \forall \mathcal{N} \subset \mathcal{M} \subset \mathcal{E}_k^{RT},$ $\forall m \in \mathcal{E}_k^{RT}$, then the result of OTAP is at least better than $1 - 1/e$ of an optimal assignment. That is, $Q_\theta^N(s_k, \mathcal{M}_\theta^{(N)}) \geq Q^{|\mathcal{M}^*|}(s_k, \mathcal{M}^*)(1 - 1/e)$. See Appendix H and K for the proofs.*

### 6.2 Performance bound of AFQI

AFQI seeks to find $\theta$ that minimizes $E_{(s_k, a_k, r_k, s_{k+1}) \sim D}\left[Q_\theta\left(s_k, a_k\right) - [r\left(s_k, a_k\right) + \gamma Q_\theta\left(s_{k+1}, \pi_{Q_\theta}\left(s_{k+1}\right)\right)]\right]$. Here, we use OTAP, denoted as $\pi_{Q_\theta}$, instead of $\max_a Q_\theta\left(s_k, a\right)$ which is used in general fitted-Q iteration. As we have seen in section 5.1, OTAP replaces the *max* operation with the auction algorithm with a provable performance bound compared with the *max* operation. *Lemma 3* allows us to use this performance bound to obtain a performance assurance on AFQI compared with FQI. We only write an abbreviated version of the statement for brevity. The formal description of conditions and the proof is provided in Appendix I.

**_Lemma 3._** *Kang & Kumar (2021)* Suppose that a $1 - 1/r$ approximation algorithm is substituted for the $\max$ operation in FQI. Then, the corresponding new Fitted Q-iteration's performance is at least $1 - 1/r$ optimal.

**_Corollary 1._** AFQI achieves at least $1 - 1/e$ performance compared with the optimal policy.

Table 1: Performance test (50 trials of training for each case)

| Reward | Environment | Baseline | Testing size : Robot (R) / Task (T) | | | | | | |
|---|---|---|---|---|---|---|---|---|---|
| | | | 2R/20T | 3R/20T | 3R/30T | 5R/30T | 5R/40T | 8R/40T | 8R/50T |
| Linear | Deterministic | Optimal | 98.31 (±4.23) | 97.50 (±4.71) | 97.80 (±5.14) | 95.35 (±5.28) | 96.99 (±5.42) | 96.11 (±4.56) | 96.85 (±3.40) |
| | | Ekisi et al. | 99.86 (±3.24) | 97.50 (±2.65) | 118.33 (±2.84) | 110.42 (±2.97) | 105.14 (±3.78) | 104.63 (±2.50) | 120.16 (±3.94) |
| | | SGA | 137.3 (±5.65) | 120.6 (±5.03) | 129.7 (±5.54) | 110.4 (±4.34) | 123.0 (±4.97) | 119.9 (±4.74) | 119.8 (±5.84) |
| | Stochastic | SGA | 130.9 (±4.02) | 115.7 (±4.03) | 122.8 (±5.21) | 115.6 (±6.23) | 122.3 (±4.94) | 113.3 (±5.53) | 115.9 (±4.08) |
| Nonlinear | Deterministic | SGA | 111.5 (±3.71) | 118.1 (±5.56) | 118.0 (±5.09) | 110.9 (±4.64) | 118.7 (±5.23) | 111.2 (±5.38) | 112.6 (±5.07) |
| | Stochastic | SGA | 110.8 (±5.17) | 117.4 (±6.22) | 119.7 (±4.48) | 111.9 (±4.70) | 120.0 (±6.38) | 110.4 (±5.14) | 112.4 (±5.30) |

Table 2: Transferability test (linear & deterministic env, standard dev. provided in the appendix)

| Training size | Testing size : Robot (R) / Task (T) | | | | | | |
|---|---|---|---|---|---|---|---|
| #R/ #T | 2R/20T | 3R/20T | 3R/30T | 5R/30T | 5R/40T | 8R/40T | 8R/50T |
| 2R/20T | 98.31 (±4.23) | 93.61(±4.98) | 97.31 (±4.25) | 92.16 (±3.49) | 92.83(±4.25) | 90.94(±3.98) | 93.44 (±4.02) |
| 3R/20T | 95.98(±4.75) | 97.50(±3.71) | 96.11(±3.63) | 93.64(±4.54) | 91.75(±5.71) | 91.60(±5.03) | 92.77(±4.74) |
| 3R/30T | 94.16(±4.97) | 96.17(±4.22) | 97.80(±5.14) | 94.79(±3.53) | 93.19(±3.78) | 93.14(±4.50) | 93.28(±3.99) |
| 5R/30T | 97.83(±3.11) | 94.89(±4.43) | 96.43(±4.23) | 95.35±5.28) | 93.28(±4.18) | 92.63(±5.07) | 92.40(±4.10) |
| 5R/40T | 97.39(±4.65) | 94.69(±4.01) | 95.22(±4.88) | 93.15(±5.09) | 96.99±4.42) | 94.96(±3.94) | 93.65 (±5.66) |
| 8R/40T | 95.44(±4.32) | 94.43(±4.88) | 93.48(±4.37) | 93.93(±5.05) | 96.41(±3.96) | 96.11±4.56) | 95.24(±4.44) |
| 8R/50T | 95.69(±3.18) | 96.68(±2.81) | 97.35(±4.20) | 94.02(±2.69) | 94.50(±4.44) | 94.86(±3.26) | 96.85±3.40) |

Table 3: Training complexity (mean of 20 trials of training, linear & deterministic env.)

| Linear & Deterministic | Testing size : Robot (R) / Task (T) | | | | | | |
|---|---|---|---|---|---|---|---|
| | 2R/20T | 3R/20T | 3R/30T | 5R/30T | 5R/40T | 8R/40T | 8R/50T |
| Performance with full training | 98.31 | 97.50 | 97.80 | 95.35 | 96.99 | 96.11 | 96.85 |
| Training time for 93% optimality | 19261.2 | 61034.0 | 99032.7 | 48675.3 | 48217.5 | 45360.0 | 47244.2 |

## 6.3 Experiment settings

In the main text, we focus on discrete-time & discrete-state (DTDS) MRRC problems with deterministic and stochastic task completion times. For CTCS deterministic problems with real-world datasets, see IPMS (Appendix A) and mTSP (Appendix B).

**Environment.** Since there is no standard dataset for MRRC problems, we used the complex maze-like environment generator of Neller et al. (2010) (code provided in Appendix 10). This complex maze mimics the complex road layout of a city and random traffic, inducing nontrivial task completion times. See the leftmost image of Figure 2 and the supplementary video. We randomly generated a new maze for every training and testing experiment with randomly chosen initial task/robot locations. To generate the task completion times, Dijkstra's algorithm and dynamic programming were used for deterministic and stochastic environments, respectively.

In the stochastic environment, a robot makes its intended move with a certain probability. (Cells with a dot: success with 55%, every other direction with 15% each. Cells without a dot: 70% and 10%, respectively.) A task is considered served when a robot reaches it. We consider two reward rules: linearly decaying rewards $f(age) = \max\{200 - age, 0\}$ and nonlinearly decaying rewards $f(age) = \lambda^{age}$ with $\lambda = 0.99$, where $age$ is the task age when served. The initial age of tasks are uniformly distributed in the interval $[0, 100]$.

**Baselines.** For deterministic environments with linear rewards, where the corresponding MRRC can be formulated as a mixed-integer linear program (MILP), we consider the following two baselines:
- *Optimal*: Gurobi Gurobi Optimization (2019), an off-the-self the optimization solver for MILP, was used to solve the problems with 60-min time limit.
- *Ekici et al.*: Ekici & Retharekar (2013), the most up-to-date heuristic for solving MRRC in the Operations Research community, was used the problems.

For stochastic environments or exponential rewards, to our knowledge, there is no literature addressing MRRC with. Thus, we construct an indirect baseline:
- *Sequential Greedy Algorithm (SGA)*: a general-purpose multi-robot task allocation algorithm called SGA Han-Lim Choi et al. (2009).

The performance measure we used is $\rho = \frac{\text{Rewards collected by the proposed method}}{\text{Reward collected by the baseline}}$. Thus, the value of $\rho$ greater than 100% indicates the proposed method collects more reward than the corresponding baseline algorithm. Note that $\rho$ against Optimal is always lower than 100%.

Note that we cannot provide other reinforcement learning-based heuristics as additional baselines since, to the best of our knowledge (for the class of NP-hard multi-robot/machine scheduling problems with decaying rewards), this paper is the first to propose a reinforcement learning-based heuristic.

### 6.4 Performance test.

Performance was tested under four environments: deterministic/linear rewards, deterministic/nonlinear rewards, stochastic/linear rewards, stochastic/nonlinear rewards. See Table 1. Our method achieves near-optimality for linear/deterministic rewards with 3% fewer rewards than *optimal* on average. The standard deviation for $\rho$ is provided in parentheses. For other environments, we see that the %SGA ratio for linear/deterministic is well maintained. Due to dataset generation's dynamic programming computation complexity, we only consider 8 robots/50 tasks at maximum. We considered larger size problems in IPMS experiments discussed in Appendix A.

### 6.5 Transferability test.

Table 2 provides comprehensive transferability test results. The rows indicate training conditions, while the columns indicate testing conditions. The results in the diagonal cells in red (cells with the same training size and testing size) serve as baselines (direct testing). The results in the off-diagonal show the results for the transferability testing and demonstrate how the algorithms trained with different problem sizes perform well on test problems (zero-shot transfer). We can see that lower-direction transfer tests (trained with larger size problems and tested with smaller size problems) show only a small loss in performance. For upper-direction transfer tests (trained with smaller size problems and tested with larger size problems), the loss was up to 4 percent.

### 6.6 Scalability analysis.

For training complexity, we measured the training time required to achieve 93% optimality considering a deterministic environment with linear rewards. Table 4 shows that training time may not necessarily increase as problem size gets larger, while the performance is fairly maintained.

MRRC can be formulated as a semi-MDP (SMDP) based multi-robot planning problem (e.g., Omidshafiei et al. (2017)). This problem's complexity with $R$ robots and $T$ tasks and maximum H time horizon is $O((R!/T!(R-T)!)^H)$. In our proposed method, this complexity is addressed by a combination of two complexities: computational complexity and training complexity. For computational complexity of joint assignment decision at each timestep is $O(|R||T|^3)$. See Appendix L for details.

## 7 Concluding Remarks

In this paper, we addressed the challenge of developing a near-optimal learning-based method for solving NP-hard multi-robot/machine scheduling problems. We developed a theory of mean-field inference for scheduling problems and a corresponding theoretically justified GNN method to precisely infer the Q-function. We addressed the scalability issue of Fitted Q-Iteration methods for multi-robot/machine scheduling problems by providing a polynomial-time algorithm with a provable performance guarantee. Simulation results demonstrate the effectiveness of the our methods.

**Acknowledgement**

Jinkyoo Park was supported by Institute of Information & communications Technology Planning Evaluation (IITP) grant funded by the Korea government(MSIT)(2022-0-01032, Development of Collective Collaboration Intelligence Framework for Internet of Autonomous Things).

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
