# OpenReview forum: "Learning NP-Hard Multi-Agent Assignment Planning using GNN: Inference on a Random Graph and Provable Auction-Fitted Q-learning"
_NeurIPS.cc/2022/Conference — NeurIPS 2022 Accept_

### Official Review · Reviewer_JnjH · 2022-07-08

**Rating:** 4
**Confidence:** 1
**Soundness:** 3 good
**Presentation:** 1 poor
**Contribution:** 3 good

**Summary:**

The paper describes a method to approximately solve NP-hard problem MRRC with RL algorithms in polynomial time with a guarantee on the result. Related theorems are proved and computational examples are made.

**Questions:**

Why the MRRC problem was chosen? Is it a popular studying topic?

In line 88 the set $\{1,\ldots,N\}$ is not indexed. $N$ must depend on $k$, doesn’t it?

In line 90 $E_k^{TT}$ is introduced. We think that clarification is needed on why these edges are needed and how they are computed and used.

In line 126 there’s a typo. Probably not “task locations”, but “robot locations”.

In line 339 double “the”.

The authors have found a problem which is solved well by their method, but do there exist NP-hard problems for which the paper’s method doesn’t work well? It seems the authors didn’t try to find such problems.


**Limitations:**

The article does not have potentially negative societal impact.

**Strengths And Weaknesses:**

I am not specialists in the areas of NP-hard problems and Bayesian networks so it is hard for me to assess the claims made by authors in these areas. Perhaps, there was a mistake to make me a reviewer of the paper. Nonetheless, in my poinion, the work seems to be original and fairly significant.

There are many typos in the work: missing parentheses, extra whitespaces in the center of words, incorrect use of plural form of words.

---

> ### Author Response · Authors · 2022-08-02
> **Thank you for your positive and constructive feeback.**
>
> **Weakness 1. Why the MRRC problem was chosen? Is it a popular studying topic?**
>
> We chose this topic because scheduling multiple agents for the tasks with time-dependent rewards is an important problem that can model many real-world problems.
>
> The particular MRRC problem was chosen because it is the simplest yet nontrivial problem that can illustrate to the community the possibility of addressing time-dependent rewards.
>
> As no prior work we know in the reinforcement learning community has newly formulated/chosen an existing multi-agent scheduling problem with time-dependent rewards and addressed it, we chose existing problems in the Operations Research community as Dai et al. 2017 did.
>
> **Weakness 2. Do there exist NP-hard problems for which the paper’s method doesn’t work well?**
>
> Thank you for your comment. We tried Identical parallel machine scheduling problems (IPMS) and the min-max multiple Traveling salesman problems (minmax-mTSP) in the Appendix and we saw comparable success for those applications also.
>
> In terms of future directions, we have not yet tried the same problem with heterogeneous agents, where the agent's speeds are different.

---

### Official Review · Reviewer_VzR4 · 2022-07-10

**Rating:** 4
**Confidence:** 2
**Soundness:** 3 good
**Presentation:** 1 poor
**Contribution:** 2 fair

**Summary:**

This paper tackles the NP-hard problem of assigning multiple agents (machines, robots etc.) to multiple tasks, where the tasks have decaying rewards over time. The solution taken is to model the problem as a graph, learn a Q function  using a graph neural network which can be used to approximate the value of each possible assignment in a problem, then use the Q function to score bids in an auction mechanism to determine the final allocation. The approach is evaluated against appropriate baselines on range of large problem instances from the same problem class.

**Questions:**

1. Why only scale to 8 robots? I would definitely like to have seen larger numbers to show the power of this vs baselines.
2. Can you give a formal statement of the (optimisation/assignment) problem you're trying to solve?
3. Is the mechanism described in 4.5 used at train or inference time?
4. What is the difference between the mechanism in 4.5 a Single Sequential Item auctioning?
5. What's the baseline for the transferability results?
6. Why 93% optimality? How is this determined?


**Limitations:**

Limitations are not obviously discussed.

**Strengths And Weaknesses:**

Whilst I work on task allocation, I am not too familiar with GNNs and random graphs, so I cannot accurately comment on all parts of this paper.

I will start by saying that this is an interesting class of problem and one that is highly relevant in real-world applications. The problem sizes in this paper are interesting but could have a larger number of robots.

In general I am excited about this paper, but think the content is let down by the presentation.  You have a lot to explain in a small number of pages. It feels like you're trying to write a journal paper in conference format (with a lot of content in the appendix).

The related work is very light. You have only focussed on very directly related work, but isn't there a lot more in the space of trying to solve combinatorial optimisation problems via deep RL techniques?

Despite the level of formality introduced, there isn't really a formal statement of the problem being solved.

It's hard to grasp the experimental domain without an image in the main paper.

What\s the base;om

The mapping from MRCC states/actions to the random PGM is very interesting, but is not explained clearly enough. I don't really follow why it's called a *random* PGM, and found it unclear whether you're discussing the PGM or a realisation of it in places.

The construction of a state in the MRCC problem as a PGM is clear, but then the structure2vec part is hard to follow. This could just be down to my lack of background in this area though.

Following this, I generally struggle to grasp the overall structure of your algorithm. I think I understand correctly that instead of learning a Q function with a max in it following the standard Bellman update, you replace the max with a the auction mechanism. But I don't quite follow whether this is at training time or test time. I think it's the former, but I struggle to imagine the training regime (again this could be my unfamiliarity with the application of GNNs/learning methods in this space).

Section 4.5 seems to be describing a Single Sequential Item auction. Is that correct?

The results in Table 1 look strong and are the main reason for my excitement about the paper. Table 2 is less clear. Is there some trend? Table 3 is even less clear. Why 93%? How does this relate to "full training"?

I'm sorry this isn't a more helpful technical review. There's clearly strong work here, but I think you need to rethink your overall story and presentation so that your audience can follow along a little easier.

---

> ### Author Response · Authors · 2022-08-02
> **We would be appreciated it if you could check the rebuttal revision of the paper.**
>
> **Weakness 1. The related work is very light. You have only focused on very directly related work.**
>
> We have conducted a more extensive literature review to summarize the recent research trend in applying reinforcement learning approaches to vehicle routing problems. The following sub-section will be included in the camera-ready version manuscript if the paper is accepted.
>
> $\textbf{Reinforcement Learning based Vehicle Routing Problems}$. RL approaches to solving vehicle routing problems can be categorized into three: (1) the improvement heuristics that learn an operator can iteratively improve the entire routing plans until there is no improvement is made [1,2,3,4], (2) the construction heuristics that learn a policy that sequentially make a single routing action given the partial solution (state) [5,6,7,8], and (3) the hybrid approaches that mix these two approaches [9,10,11,12]. These RL approaches have mainly considered a single-agent routing problem. Although these methods solve CVRP with multiple vehicles, they solve this problem from a single-agent perspective. In addition, most of these approaches consider the static reward function setting. On the contrary, our study explicitly considers multi-vehicle interaction while considering the time-varying reward, which is a more realistic routing problem setting.
>
>
>
>
>
>
>
> [1] Wu, Y., Song, W., Cao, Z., Zhang, J., and Lim, A. "Learning improvement heuristics for solving routing problems," IEEE Transactions on Neural Networks and Learning Systems, 2021.
>
> [2] Chen, X. and Tian, Y. "Learning to perform local rewriting for combinatorial optimization," Neural Information Processing Systems, 2019.
>
> [3] Lu, H., Zhang, X., and Yang, S. "A learning-based iterative method for solving vehicle routing problems," International Conference on Learning Representations, 2020.
>
> [4] Kim, M., Park, J., Kim, J. "Learning collaborative policies to solve NP-hard routing problems," Advances in Neural Information Processing Systems, 2021.
>
> [5] Bello, I., Pham, H., Le, Q. V., Norouzi, M., and Bengio, S. "Neural combinatorial optimization with reinforcement learning," arXiv preprint arXiv:1611.09940, 2016.
>
> [6] Nazari, M., Oroojlooy, A., Snyder, L., and Takác, M. "Reinforcement learning for solving the vehicle
> 464 routing problem," Advances in Neural Information Processing Systems, 2018.
>
> [7] Kool, W., Van Hoof, H., and Welling, M. "Attention, learn to solve routing problems," International Conference on Learning Representations, 2019.
>
> [8] Dai, H., Khalil, E. B., Zhang, Y., Dilkina, B., and Song, L. "learning combinatorial optimization algorithms over graphs," Advances in Neural Information Processing Systems, 2017.
>
> [9] Joshi, C. K., Cappart, Q., Rousseau, L.-M., Laurent, T., and Bresson, X. "Learning tsp requires rethinking generalization," arXiv:2006.07054, 2021.
>
> [10] Fu, Z.-H., Qiu, K.-B., and Zha, H. "Generalize a small pre-trained model to arbitrarily large tsp instances," Association for the Advancement of Artificial Intelligence, 2021.
>
> [11] Kool, W., van Hoof, H., Gromicho, J., and Welling, M. "Deep policy dynamic programming for vehicle routing problems," arXiv:2102.11756, 2021.
>
> [12] Ahn, S., Seo, Y., and Shin, J. "Learning what to defer for maximum independent sets," International Conference on Machine Learning, 2020.

---

> > ### Author Response · Authors · 2022-08-02
> > **(Continued)**
> >
> > **Weakness 2. It's hard to grasp the experimental domain without an image in the main paper.**
> >
> > Due to lack of space, in the previously submitted version, we could not include Figure 2 describing the overall procedure of the method in the main text. Currently, this figure is in the last section of supplementary material. We believe that Figure 2 can help you understand the proposed method and the experiment setups. We will include this main figure in the camera-ready version if the paper is accepted.
> >
> > **Weakness 3. I don't really follow why it's called a random PGM, and found it unclear whether you're discussing the PGM or a realization of it in places.**
> >
> > We included the following explanation in the rebuttal revision of the paper:
> >
> > "We call it random PGM because we model `A distribution over all possible PGMs.' That is, a PGM will be realized among many possible PGMs according to the distribution. The intuition goes like this. Even when you know the entire set of routes of all the agents, uncertainty in traveling time causes rewards associated with that set of routes to be uncertain. A Bayesian Network can be used to model the rewards. But what if the situation is worse and you are not even certain about the routes of all agents? This case is modeled by random PGM: A route is realized probabilistically among many possible routes."
> >
> >
> > **Weakness 4. I don't quite follow whether this is at training time or test time.**
> >
> > Thank you for your feedback. We wrote more clearly about this in the rebuttal revision of the paper.. The auction mechanism happens either during the training or testing.
> >
> >
> > **Weakness 5. Table 2 is less clear.**
> >
> > We agree that this shape of the table may frustrate the reader except for the daily users of the transferability concept in this particular narrow area. In Table 2, the baselines are the diagonal ones, where the training environment and testing environments are the same. The concept of transferability is about whether the testing result will be still good even when the test environment is not the same as the training environment.
> >
> > **Weakness 6. Why 93percent ?**
> >
> > One approach to measure scalability in data efficiency is to do like we did, and the other is to wait until the performance is stabilized. We follow the former, which is typical because it is a more conservative comparison in this area. I will explain why it is more conservative.
> >
> > Most algorithms in this area, including ours, provides a higher stabilized performance for a smaller system. For example, for a smaller system, an algorithm might be able to reach 98\% stabilized performance at around 40000 training steps and 93\% performance at around 10000 training steps. for a larger system, algorithm reaches 95\% stabilized performance at around 40000 training steps and 93\% performance at around 20000 training steps. If we only compare the iterations it took to reach stabilized performance and say that our algorithm is perfectly scalable (because for both small and large systems it took 40000 iterations to stabilize), we are exaggerating our algorithm's scalability.

---

> > > ### Author Response · Authors · 2022-08-02
> > > **(Continued)**
> > >
> > > **Question 1. Why 8 robots?**
> > >
> > > We agree that showing the algorithm's performance for larger robots could illustrate the real-world applicability of our algorithm. Indeed, the complexity of our algorithm is $O\left(|R||T|^{3}\right)$ (see Appendix L), where $R$ is the robot set and $T$ is the task set. This means that increasing the number of agents would not be hard. However, to make the problem nontrivial we should also increase the number of tasks and this is computationally burdening ($50^3$ and $100^3$ was a huge difference for a personal PC). This was the reason why we considered 8 robots at maximum.
> > >
> > >
> > > **Question 2. Can you give the formal Statement of the problem?**
> > >
> > > The target problem is cast as a reinforcement learning problem for solving the target MDP problem as:
> > >
> > > "Denote the state at $t_k$ as $s_{t_k}$. A stationary policy $\pi$ is a function that maps current state $s$ into current action $a$.The state at the next decision epoch $s^{\prime}\sim P(s, a)$. The reward function is $R(s, a, s^{\prime})$. Our well defined objective is to seek a stationary policy $\pi$ to maximize $Q^{\pi}(s, a) =:E_{P, \pi}[\sum_{k=0}^\infty R(s_{t_k}, a_{t_k}, s_{t_{k+1}}) | s_{t_{0}}=s, a_{t_0}=a]  $."
> > >
> > > A fundamental result in dynamic programming and offline reinforcement learning  [reference: Kumar and Variaya 2015 book] is that the problem of finding a good stationary policy for such sequential decision-making problems can be solved instead by solving the problem of precisely estimating $Q^{\pi}(s, a)$ of a policy ${\pi}$ for each $(s, a)$. Note that the system dynamics ($P(s, a)$) or reward function may not be needed to solve this estimation problem. In this context, when formulating a sequential decision-making problem, it remains to choose the state-action representation $(s, a)$. A good choice here will enable precise inference of $Q^{\pi}(s, a)$.
> > >
> > > We will include the above description in the camera-ready version manuscript if the paper is accepted.
> > >
> > >
> > > **Question 3. Is the mechanism described in 4.5 used at train or inference time?**
> > >
> > > The mechanism is used for both training and testing. We added a sentence saying that.
> > >
> > > **Question 4. What is the difference between the mechanism in 4.5 a Single Sequential Item auctioning?**
> > >
> > > The auction introduced in this paper is designed to be a bit more conservative to address the non-linearity of the value function. Let's consider a simple example of a sub-procedure of the auction. Suppose that Agent 1 bids 80 for task $A$ and Agent 2 bids 50 for task $B$. There is no conflict among agents and obviously, Agent 1 must get task $A$. But for Agent 2 we don't assign task $B$. What can happen is that, Task $A$ and Task $B$ may be very close, and it can be that Agent 2's value on Task $B$ can decrease from 50 to 10 knowing that Agent 1 gets Task $A$. Agent $2$ might want Task $C$ more now.
> > >
> > > **Question 5. ``What's the baseline for the transferability results?**
> > >
> > > We answer this question in Weakness 5, so please refer to it.
> > >
> > > **Question 6.Why 93 percent optimality?**
> > >
> > > We answer this question in Weakness 6, so please refer to it.

---

> > > > ### Comment · Reviewer_VzR4 · 2022-08-08
> > > > **Rebuttal feedback**
> > > >
> > > > Thanks for your time writing this response. This is reasonably clear and I feel I better understand the paper now.
> > > >
> > > > I'm curious about the train vs test usage of the auction. If you are learning from the outcomes of the auction mechanism, do you really need to re-run auctions at test time? I appreciate that there is a huge combinatorial space here, which could be why you need to return the search-life functionalities of the auction.
> > > >
> > > > Your example for your algorithm being not SSI describes exactly how SSI functions. That's the key part of the "Single" element - only one item is assigned per round, allowing for these kinds of changes in preferences to take place.
> > > >
> > > > You have given a formal statement of the general RL problem, not the TA. I understand you need to maximise reward in RL, but what is the objective function/model for the TA problem that you translate into the RL problem?
> > > >
> > > > The scalability issue is an interesting one. You easily scale with the number of agents, but the problem only gets interesting/relevant as you scale tasks. Since your auction is at both train and test time, it seems like you'll have problems here.

---

> > > > > ### Author Response · Authors · 2022-08-08
> > > > > **Thank you for giving us an opportunity to interact.**
> > > > >
> > > > > Thank you for giving us an opportunity to interact.
> > > > >
> > > > > $\textbf{Answer to:}$ $\textit{Do you really need to re-run auctions at test time?}$
> > > > >
> > > > > Your intuition is correct: for the case when you don't need a very fast assignment or the number of robots and tasks are small, you don't need auctioning - you can simply compare all possible assignments of each timestep. In the case when a real-time fast assignment decision is required or the number of tasks is large, auctioning may significantly help the computational burden - without auction, you need $O\left((R ! / T !(R-T) !)\right)$ computations. For example, see below for the example of Boston's Uber, where $O(RT^3)$ computational complexity of the auction makes the real-time scheduling reasonable.
> > > > >
> > > > > $\textbf{Answer to:}$ $\textit{Your example for your algorithm being not SSI describes exactly how SSI functions.}$
> > > > >
> > > > > Thank you for improving our knowledge and the paper. We checked that what you are saying is right. We added the following sentence to the paper's line 275-276:
> > > > >
> > > > > This auction follows the spirit of Sequential Single Item (SSI) auctioning [1]; each iteration adds one robot-task assignment to construct a full joint assignment.
> > > > >
> > > > > $\textbf{Answer to:}$ $\textit{That is the objective function/model for the TA problem?}$
> > > > >
> > > > > The objective function we wrote is in terms of the stochastic optimization problem [3]. Just in case you are interested in the deterministic optimization problem's objective, here is the formulation we can use, suggested by [4]:
> > > > >
> > > > > $\max \sum_{i=1}^{|A|}(\sum_{j=1}^{|T|}c_{i j}(x_i,p_i)x_{ij} )$
> > > > >
> > > > > subject to $\sum_{i=1}^{|A|} x_{i j} \leq 1, \forall j \in T$
> > > > >
> > > > > where $A$ is the agent set, $T$ is the task set, $x_{i j}=1$ if agent $i$ is assigned to task $j$ and $0$ otherwise, and $x_{i} \in {{0,1}}^{|T|}$ is a vector whose $j$-th element is $x_{i j}$. The vector $p_{i} \in(T \cup\{\emptyset\})^{|A|}$ represents a sequence of tasks in the order of agent $i$ performs; its $k$-th element is $j$ if agent $i$ conducts $j$ at the $k$-th position along the path. $c_{i j}\left(x_{i}, p_{i}\right)$ is a function that is derived by the following information: predetermined reward function introduced in the paper, initial positions of agents and tasks, and the method of routing.
> > > > >
> > > > > $\textbf{Answer to:}$ $\textit{Since your auction is at both train and test time,}$
> > > > > $\textit{it seems like you'll have problems here.}$
> > > > >
> > > > > It is right that $O(RT^3)$ still gives a challenging computational burden for a Ghz-level personal computer without parallel computing. This problem, however, is less of a concern for the commerce-scale business, such as Uber or Lyft, because of two reasons:
> > > > >
> > > > > 1) You can decompose the scheduling problem into city-level scale problems, as Uber and Lyft do in the real world (for example, at Uber, each city's operations office has its own dynamic pricing policy and solves the city-size problem)
> > > > >
> > > > > 2) If the number of drivers is 1000 and the number of riders is 1000 (which is a reasonable number even in Boston, considering that most Uber drivers are part-time drivers- see [2]), the $10^{12}$ per second computation requirement is a very easy number for a modern parallel computing servers.
> > > > >
> > > > > [1] Koenig, Sven, et al. "The power of sequential single-item auctions for agent coordination." AAAI. Vol. 2006. 2006.
> > > > >
> > > > > [2] https://www.bostonherald.com/2015/01/22/data-10000-uber-cars-dwarf-licensed-taxis-in-boston/
> > > > >
> > > > > [3] Kumar, P. R., & Varaiya, P. (2015). Stochastic systems: Estimation, identification, and adaptive control. Society for industrial and applied mathematics.
> > > > >
> > > > > [4] Choi, Han-Lim, Luc Brunet, and Jonathan P. How. "Consensus-based decentralized auctions for robust task allocation." IEEE transactions on robotics 25.4 (2009): 912-926.

---

> > > > > > ### Author Response · Authors · 2022-08-09
> > > > > > **Thank you for giving us an opportunity to interact (2)**
> > > > > >
> > > > > > We answered additional questions from reviewers. We would appreciate it if the reviewer could check our responses. We hope that our response resolves the reviewer's concern and helps to make the final decision.

---

> ### Author Response · Authors · 2022-08-08
> **Thank you for giving us an opportunity to interact.**
>
> Thank you for giving us an opportunity to interact.
>
> $\textbf{Answer to:}$ $\textit{Do you really need to re-run auctions at test time?}$
>
> Your intuition is correct: for the case when you don't need a very fast assignment or the number of robots and tasks are small, you don't need auctioning - you can simply compare all possible assignments of each timestep. In the case when a real-time fast assignment decision is required or the number of tasks is large, auctioning may significantly help the computational burden - without auction, you need $O\left((R ! / T !(R-T) !)\right)$ computations. For example, see below for the example of Boston's Uber, where $O(RT^3)$ computational complexity of the auction makes the real-time scheduling reasonable.
>
> $\textbf{Answer to:}$ $\textit{Your example for your algorithm being not SSI describes exactly how SSI functions.}$
>
> Thank you for improving our knowledge and the paper. We checked that what you are saying is right. We added the following sentence to the paper's line 275-276:
>
> This auction follows the spirit of Sequential Single Item (SSI) auctioning [1]; each iteration adds one robot-task assignment to construct a full joint assignment.
>
> $\textbf{Answer to:}$ $\textit{That is the objective function/model for the TA problem?}$
>
> The objective function we wrote is in terms of the stochastic optimization problem [3]. Just in case you are interested in the deterministic optimization problem's objective, here is the formulation we can use, suggested by [4]:
>
> $\max \sum_{i=1}^{|A|}(\sum_{j=1}^{|T|}c_{i j}(x_i,p_i)x_{ij} )$
>
> subject to $\sum_{i=1}^{|A|} x_{i j} \leq 1, \forall j \in T$
>
> where $A$ is the agent set, $T$ is the task set, $x_{i j}=1$ if agent $i$ is assigned to task $j$ and $0$ otherwise, and $x_{i} \in {{0,1}}^{|T|}$ is a vector whose $j$-th element is $x_{i j}$. The vector $p_{i} \in(T \cup\{\emptyset\})^{|A|}$ represents a sequence of tasks in the order of agent $i$ performs; its $k$-th element is $j$ if agent $i$ conducts $j$ at the $k$-th position along the path. $c_{i j}\left(x_{i}, p_{i}\right)$ is a function that is derived by the following information: predetermined reward function introduced in the paper, initial positions of agents and tasks, and the method of routing.
>
> $\textbf{Answer to:}$ $\textit{Since your auction is at both train and test time,}$
> $\textit{it seems like you'll have problems here.}$
>
> It is right that $O(RT^3)$ still gives a challenging computational burden for a Ghz-level personal computer without parallel computing. This problem, however, is less of a concern for the commerce-scale business, such as Uber or Lyft, because of two reasons:
>
> 1) You can decompose the scheduling problem into city-level scale problems, as Uber and Lyft do in the real world (for example, at Uber, each city's operations office has its own dynamic pricing policy and solves the city-size problem)
>
> 2) If the number of drivers is 1000 and the number of riders is 1000 (which is a reasonable number even in Boston, considering that most Uber drivers are part-time drivers- see [2]), the $10^{12}$ per second computation requirement is a very easy number for a modern parallel computing servers.
>
> [1] Koenig, Sven, et al. "The power of sequential single-item auctions for agent coordination." AAAI. Vol. 2006. 2006.
>
> [2] https://www.bostonherald.com/2015/01/22/data-10000-uber-cars-dwarf-licensed-taxis-in-boston/
>
> [3] Kumar, P. R., & Varaiya, P. (2015). Stochastic systems: Estimation, identification, and adaptive control. Society for industrial and applied mathematics.
>
> [4] Choi, Han-Lim, Luc Brunet, and Jonathan P. How. "Consensus-based decentralized auctions for robust task allocation." IEEE transactions on robotics 25.4 (2009): 912-926.

---

### Official Review · Reviewer_99dG · 2022-07-23

**Rating:** 6
**Confidence:** 3
**Soundness:** 3 good
**Presentation:** 2 fair
**Contribution:** 3 good

**Summary:**

This paper proposes a centralized learning-based framework for solving NP-hard multi-agent multi-task scheduling problems with time-dependent rewards, which can achieve near-optimal performance. The proposed learning framework involves a Q-value estimator based on random graph embeddings for state-joint assignment pairs and an order-transferability-enabled auction policy (OTAP) to select joint assignment action while avoiding the exponential growth in the joint action space. OTAP is finally incorporated into the fitted-Q-iteration algorithm, called Auction-Fitted-Q-Iteration (AFQI), which is proved to generate a policy with polynomial-time and achieve near-optimal performance.  Evaluation is conducted under a maze-like environment with both deterministic and stochastic dynamics. The performance of the proposed approach is compared with three baselines.


**Questions:**

N/A

**Strengths And Weaknesses:**

Strengths:

This work is novel as this is the first work to solve NP-hard multi-agent scheduling problems with decaying rewards via reinforcement learning. This paper not only addresses a scalability issue on exponential joint action space in centralized control by developing AFQI with polynomial computation time but also provides sound theoretical analysis. Empirical results show the near-optimal performance achieved by the proposed method and successfully demonstrate the advantage of the proposed method over the baselines in terms of the scalability to a large number of agents and tasks and the transferability to new tasks.

Weaknesses:

The paper can be further strengthened by having problems where agents could have different traveling speeds or maybe involve heterogeneous agents such as having different speeds to finish the same task.

---

> ### Author Response · Authors · 2022-08-02
> **Thank you for your review and positive feedback.**
>
> Thank you for your review and positive feedback. We agree that addressing heterogeneity in the ability among agents is certainly a challenging but important future direction.

---

> > ### Comment · Reviewer_99dG · 2022-08-08
> > **Reviewer Response**
> >
> > Thank you for the response. After checking the answers to other reviewers' questions, I am convinced by the originality of this work. I would stick with my score. I agree with other reviewers' concerns about clarification, and suggest highlighting the updates using a different color in the revision, which would be easy for other reviewers to detect the changes.

---

> > > ### Author Response · Authors · 2022-08-09
> > > **Thank you for acknowledging our responses.**
> > >
> > > Thank you so much for acknowledging our responses. As suggested, the corrected part is marked with blue font in the revised manuscript. Unfortunately, due to the 9-page limit, we could not modify the manuscript much. However, we will include the literature review part and the overall figure showing the flow of the entire algorithm if the paper is accepted (we can use ten pages).

---

### Author Response · Authors · 2022-08-09
**Thank you for your hard work and efforts.**

As we are approaching the end of this rebuttal period, we would like to express our sincere thanks to all reviewers for participating discussion actively. We appreciate all their hard work and efforts in providing comments and suggestions. This constructive discussion indeed helps us improve the quality of our submission. We sincerely wish the subsequent discussions proceed smoothly and fruitfully, and we thank the ACs, SACs, and PCs for their continuous efforts in reviewing our paper.

---

### Meta-Review · Area_Chair_no2E · 2022-08-22

**Recommendation:** Accept
**Confidence:** Less certain

**Metareview:**

The approach is novel and significant but there are concerns about the presentation in the paper. The authors should thoroughly update the paper to clarify (formally) the problem being solved and provide more details about the method. The related work should also be updated to be more extensive.

**Award:**

No

---

### Decision · Program_Chairs · 2022-09-14

Accept